# Carboplatin and Etoposide for the Treatment of Metastatic Prostate Cancer with or without Neuroendocrine Features: A French Single-Center Experience

**DOI:** 10.3390/cancers16020280

**Published:** 2024-01-09

**Authors:** Jérémy Baude, Julie Niogret, Pierre Jacob, Florian Bardet, Isabelle Desmoulins, Sylvie Zanetta, Courèche Kaderbhai, Loïck Galland, Didier Mayeur, Benjamin Delattre, Luc Cormier, Sylvain Ladoire

**Affiliations:** 1Department of Radiation Oncology, Georges-François Leclerc Cancer Center, 21000 Dijon, France; jbaude@cgfl.fr; 2Department of Medical Oncology, Georges-François Leclerc Cancer Center, 21000 Dijon, France; 3Department of Urology, University Hospital François Mitterrand, 21000 Dijon, France; florian.bardet@chu-dijon.fr (F.B.);; 4UFR des Sciences de Santé, Université of Bourgogne Franche-Comté, 21000 Dijon, France; 5Platform of Transfer in Biological Oncology, Georges-François Leclerc Cancer Center, 21000 Dijon, France; 6INSERM U1231 «Lipid, Nutrition, Cancer», 21000 Dijon, France

**Keywords:** prostate cancer, carboplatin, etoposide, neuroendocrine, survival, PSA, anaplastic, NSE, chromogranin

## Abstract

**Simple Summary:**

Chemotherapy using carboplatin and etoposide (CE) is often proposed to treat metastatic prostate cancer (mPC), both primary small-cell neuroendocrine (PSC-NE) and adenocarcinoma. We aimed to investigate the benefit of CE, especially in patients with heavily pretreated adenocarcinoma. In this retrospective series, we showed that the reports of clinical results of CE indicate that we should not mix PSC-NE and adenocarcinoma. In patients with PSC-NE, 58.8% had a radiological response to CE and median progression-free survival was 7.9 months. In pretreated patients with adenocarcinoma, the benefit/risk ratio of CE seems unfavorable with poor response and high toxicity irrespective of the elevation of neuroendocrine markers.

**Abstract:**

Background: Chemotherapy using carboplatin and etoposide (CE) is frequently pragmatically proposed to treat metastatic prostate cancer (mPC), both primary small-cell neuroendocrine (PSC-NE) carcinoma and adenocarcinoma with or without neuroendocrine (NE) marker elevation. However, the real benefit of CE is poorly reported in the recent therapeutic context. Methods: We retrospectively analyzed the efficacy and tolerance of CE chemotherapy in these three different groups of mPC patients. Efficacy endpoints included radiological response, progression-free survival (PFS), and overall survival (OS), as well as PSA response and PFS2/PFS1 ratio in patients with adenocarcinoma. Results: Sixty-nine patients were included in this single-center study (*N* = 18 with PSC-NE carcinoma and 51 with adenocarcinoma with (*N* = 18) or without (*N* = 33) NE marker elevation). Patients with adenocarcinoma were highly pretreated with next-generation hormonal agents (NHAs) and taxanes. In patients with adenocarcinoma, a PSA response ≥50% was observed in six patients (15.8%), four of whom had NE marker elevation. The radiological response was higher in PSC-NE and tended to be higher in adenocarcinoma when NE marker elevation was present. Comparing patients with adenocarcinoma with vs. without NE marker elevation, the median PFS was 3.7 and 2.1 months and the median OS was 7.7 and 4.7 months, respectively. Overall, 62.3% of patients experienced grade 3–4 adverse events (mainly hematological), and three treatment-related deaths were recorded. Conclusion: Reports of the clinical results of CE suggest that we should not mix PSC-NE and castration-resistant adenocarcinoma of the prostate. In patients with heavily pretreated adenocarcinoma, the benefit/risk ratio of CE chemotherapy seems unfavorable due to poor response and high toxicity.

## 1. Introduction

Despite numerous recent therapeutic advances, metastatic castration-resistant prostate cancer (mCRPC) remains an incurable disease and accounts for the majority of prostate cancer deaths.

The therapeutic landscape for mCRPC has broadened considerably since the mid-2000s, with overall survival (OS) benefits demonstrated for various drug strategies, including chemotherapy with taxanes (docetaxel, cabazitaxel) [1,2], next-generation hormonal agents (NHAs) [3], radioligand therapy [4,5], and PARP inhibitors (PARPi) [6] in the case of defects in the genes involved in DNA repair.

However, in everyday clinical practice, many patients have diseases that have become resistant to all of these recommended treatments, yet they remain in good general condition and require specific treatment. For these patients, there are no treatment recommendations beyond symptomatic management and palliative care, and clinicians quite frequently propose chemotherapy with available molecules that have demonstrated, most often in small trials or retrospective series, response rates (PSA or radiological) that are deemed clinically interesting. Among these classical chemotherapy agents, platinum salts, in particular carboplatin, have been extensively studied in patients with mCRPC, both as monotherapy and in combination with other molecules, especially taxanes [7,8,9]. However, in patients who have already progressed on taxanes, this type of combination is potentially of less interest, and historically, the combination of carboplatin and etoposide (CE) has often been proposed pragmatically, usually following a regimen identical to that proposed to treat advanced-stage bronchial neuroendocrine (NE) carcinomas or the rare primary small-cell neuroendocrine (PSC-NE) carcinoma of the prostate [10,11]. The main rationale for using this protocol also in primary advanced adenocarcinoma of the prostate is that tumor evolution towards an aggressive, dedifferentiated phenotype, sometimes associated with signs of NE transformation (the presence of which can be detected by blood levels of NSE or chromogranin A), is one of the adaptive phenomena of cancer cells at the castration-resistant stage [12]. This phenotype is a poor prognostic factor, associated with more frequent visceral metastases and low response rates to standard treatments [13].

Early studies, mostly carried out when docetaxel was the only chemotherapy available and before the advent of NHAs, were often conducted on cohorts of patients with heterogeneous disease. They showed discordant results as to the value of such a regimen, with PSA response rates of approximately 20%, albeit with some patients showing an objective response on imaging in addition to symptomatic benefit [10,11,14]. However, the toxicity (particularly hematological) required a careful assessment of the benefit/risk ratio for each patient. Thus, in 2019, an expert consensus considered the possibility of a platinum-based chemotherapy regimen in these advanced situations in the form of monotherapy with AUC 5–6 every 3 weeks [15].

In order to provide clinical data about the CE regimen in this indication, we report here our retrospective single-center experience of using this treatment in different clinical situations by comparing the efficacy and safety data in three different patient populations, namely: (1) patients with PSC-NE carcinoma of the prostate, (2) patients with adenocarcinoma (mCRPC) and biological signs of NE transformation (NE markers elevation), and finally, (3) patients with adenocarcinoma (mCRPC) without NE marker elevation. For the latter two populations, we included in our analysis only patients pretreated with a taxane(s) and/or one or more NHA, in order to provide clinical data in the context of current therapeutic standards.

## 2. Materials and Methods

### 2.1. Patients

Patients aged 18 years or older who received a combination of CE for the treatment of prostate cancer were included. We included patients with mCRPC or with PSC-NE prostate cancer between January 2002 and January 2021 at the Georges-François Leclerc Cancer Center (Dijon, France), a French tertiary referral center. Patients with mCRPC must have undergone androgen deprivation and have testosterone levels < 50 ng/dL. Blood levels of NE markers (neuron-specific enolase (NSE) and chromogranin A (CgA)) were routinely measured in our institution at each progression during the mCRPC clinical history. At the time of CE initiation, patients with mCRPC were considered to have NE marker elevation if NSE and/or CgA were ≥1.5 × the upper limit of normal (ULN).

In the present study, patients were classified as follows: (1) patients with histological proof of primary small-cell neuroendocrine prostate carcinoma were included in the PSC-NE group; (2) those with a primary adenocarcinoma were further dichotomized between those who presented NE marker elevation and (3) those without any NE marker elevation at the onset of CE.

### 2.2. Treatment and Follow-Up

The initial AUC of carboplatin was at the discretion of each oncologist. Etoposide was administered either orally or intravenously. Safety data were routinely collected at each cycle of chemotherapy. Radiological monitoring of the disease was performed every 3–4 months at the discretion of each oncologist by CT scan and bone scan. Since few patients in our series had radiologically measurable lesions (according to RECIST criteria), we considered the quality of the radiological response according to the conclusions of the CT scan and bone scan reports. Radiological response assessments using CT scans were based on RECIST criteria: (1) complete response (CR): disappearance of all target lesions; (2) partial response (PR): at least 30% decrease in the sum of the longest diameter of target lesions, taking as reference the baseline since treatment started; (3) progressive disease (PD): at least 20% increase in the sum of the longest diameter of target lesions, taking as reference the baseline since treatment started; and (4) stable disease (SD): neither sufficient shrinkage to qualify for PR nor sufficient increase to qualify for PD. Assessments of response by bone scan were classified as follows: (1) CR: disappearance of all bone metastasis; (2) PR: a decrease in number, extent, or intensity of bone lesions was detected; (3) PD: appearance of new bone lesion(s) and/or apparent enlargement of the bone metastases; and (4) SD: little or no change in the number, extent, or intensity of bone metastases was observed. PSA response was defined as a >30% and >50% reduction from baseline PSA.

### 2.3. Tolerance and Toxicity Evaluation

A safety evaluation was carried out systematically before each chemotherapy administration (every 3 weeks) and in the case of hospitalization for toxicity between 2 cycles of chemotherapy by each oncologist. Toxicity was assessed according to the Common Terminology Criteria for Adverse Events (CTCAE) version 5.0.

### 2.4. Statistical Analysis

For clinicopathological comparisons, continuous variables are described as mean (± standard deviation, SD) or median (with interquartile range, Q1–Q3). Categorical variables are described as the number and percentage for each modality. Percentages were calculated on complete data. Continuous variables were compared between groups using the Student *t*-test in case of normally distributed variables, and otherwise, using a Wilcoxon test. The Shapiro–Wilk test was used to check the normality of the distribution. Categorical variables were compared between groups using the chi-square or Fisher’s exact test as appropriate. The Kaplan–Meier method was used to estimate survival rates and median survival times.

Progression-free survival (PFS) was defined as the time in months between the date of the first cycle of CE chemotherapy and the date of progression or death. PFS1 was defined as the time from the start of the last treatment prior to CE chemotherapy to the date of radiological or clinical progression warranting CE chemotherapy. PFS2 was defined as the time from the start of CE chemotherapy to progression, defined as radiological or clinical progression or death from any cause. PFS2 was compared with PFS1 for each patient using the PFS2/PFS1 ratio (or growth modulation index) [16]. A PFS ratio greater than the threshold of 1.3 indicated a treatment benefit, as it signifies that PFS2 was at least 30% longer than PFS1. Overall survival (OS) was defined as the time in months between the first cycle of CE chemotherapy and death from any cause or the last follow-up. Survival curves were compared using the log-rank test.

Statistical analyses were performed using GraphPad Prism version 9.5.0. Statistical tests were two-sided and the threshold of significance was fixed at 5%.

The study was approved by an institutional review committee (ethical committee and scientific review board from Georges-François Leclerc Center) and was conducted in accordance with the Declaration of Helsinki. Patients’ non-objection to the use of their information for research purposes was collected at their admission to our institution. No patient-identifying information has been released in the present study.

## 3. Results

### 3.1. Patient Characteristics

Between 2002 and 2021, 69 patients who received the combination of carboplatin and etoposide (CE) for prostate cancer were included. The main clinical characteristics of the patients are shown in Table 1. Eighteen (26%) patients had PSC-NE. Fifty-one patients (74%) had primary adenocarcinoma of the prostate, of whom 18 (26%) presented elevated blood levels of NE markers (NSE and/or CgA) at the beginning of CE chemotherapy. The remaining 33 patients (48%) had primary adenocarcinoma without NE marker elevation at the beginning of CE treatment.

The mean age at initiation of CE chemotherapy was 69.4 ± 9.3 years. Most patients (55.1%) were in good general health (ECOG status 0–1). As expected, median PSA values were statistically significantly higher at the beginning of treatment in patients with adenocarcinoma without NE marker elevation compared to both patients with adenocarcinoma plus NE marker elevation (*p* = 0.03) and patients with PSC-NE carcinoma of the prostate (*p* < 0.01). Conversely, NE markers were higher in patients with PSC-NE carcinoma compared with the two other groups (and we also logically observed a significant difference between patients with and without NE marker elevation; *p* = 0.03).

Most patients had bone metastases at the beginning of treatment (but slightly less often in patients with PSC-NE carcinoma). Visceral metastases were present in approximately one-third of patients; liver metastases were more frequent in patients with PSC-NE carcinoma (50% of patients), and lung metastases were more frequent in patients with NE marker elevation (38.9% of patients). Conversely, lymph node metastases were more frequent in patients with primary adenocarcinoma without NE marker elevation (69.7% of patients).

### 3.2. Treatments

#### 3.2.1. Previous Treatments

The treatments previously received, prior to CE chemotherapy, are detailed in Table 1. The median number of previous lines of treatment was 3 (1–4) for the whole cohort, with significant differences between patients treated for PSC-NE carcinoma (for whom CE was given mostly as first-line treatment) and patients who received CE chemotherapy for primary adenocarcinoma with (median number: 3 (1–4)) or without (median number: 4 (4–5)) NE marker elevation (*p* = 0.01, for comparison of the last two groups).

Most patients previously received one or two NHAs. Specifically, patients with primary adenocarcinoma without NE marker elevation were most heavily pretreated with NHAs (one NHA for 24.2% and two NHAs for 69.7% of these patients). Patients with primary adenocarcinoma with NE marker elevation were significantly less often pretreated with NHAs in routine practice before CE chemotherapy (*p* < 0.01), with only 55.6% of patients having received one (27.8%) or two (27.8%) previous NHAs. As expected, patients with PSC-NE carcinoma had not received previous HNG in the vast majority of cases (94.4%). Concerning previous chemotherapies, most (96%) patients with primary adenocarcinoma received at least one chemotherapy before CE, compared to only three patients (16.7%) with PSC-NE carcinoma. In patients with primary adenocarcinoma, those with NE marker elevation received significantly fewer chemotherapy lines compared to patients without NE marker elevation, who generally previously received both docetaxel and cabazitaxel (*p* < 0.01). Individual data concerning treatments previously received by each patient in the “patients with primary adenocarcinoma” group are given in Appendix A.

#### 3.2.2. Carboplatin/Etoposide Chemotherapy

Patients received the CE association for a median of 4 (2–8) cycles in the whole cohort. Patients treated for PSC-NE carcinoma received a median of 8.5 cycles (5–12), while patients treated for primary adenocarcinoma received CE for a median of 4.5 cycles (1–8) when NE marker elevation was detectable and for a median of 3 cycles (2–4) when no NE marker was detectable. Sixty-two patients (90%) received a CE schedule given every three weeks. Etoposide was administered intravenously in 65 (94%) patients. The median initial AUC of carboplatin was 4.0 (4.0–5.0). During the treatment, 42 patients (61%) received G-CSF prophylaxis and twenty-three patients (33%) required a dose reduction.

#### 3.2.3. PSA Response in Patients Treated for Primary Adenocarcinoma

The best PSA response observed in patients treated with CE chemotherapy for adenocarcinoma is depicted for each patient in the waterfall plot shown in Figure 1. PSA evolution was documented in *N* = 38 patients (12 with and 26 without NE marker elevation). A PSA response ≥ 30% was observed in seven patients (18.4%), five of whom had NE marker elevation. A PSA response ≥ 50% was observed in six patients (15.8%), four of whom had NE marker elevation.

#### 3.2.4. Radiological Response

Forty-six patients had one or more radiological assessments after starting CE chemotherapy. The best radiological response recorded for each evaluable patient (*N* = 46 in the whole cohort, including *N* = 17 patients with PSC-NE carcinoma) is presented in Figure 2. No patient showed a complete response (CR). A partial response (PR) was observed in 58.8% of patients with PSC-NE carcinoma, in 27.4% of patients with adenocarcinoma with NE marker elevation, and in 16.7% of patients with adenocarcinoma without NE marker elevation. Considering stable disease with a partial response as a clinical benefit, clinical benefit rates were 94.1%, 63.7%, and 44.4% in the three groups, respectively.

Primary resistance to CE chemotherapy (i.e., progressive disease as best radiological evaluation) was observed in 5.9%, 36.3%, and 55.6% of the three groups, respectively. Concerning the 23 patients without radiological evaluation, CE chemotherapy was prematurely interrupted for rapid biochemical progression (*N* = 9), obvious clinical progression (*N* = 2), deteriorating health status (*N* = 3), treatment-related toxicity (*N* = 3), or death (*N* = 6). Except for one early death, all cases of premature chemotherapy interruption were observed in patients with primary adenocarcinoma.

#### 3.2.5. Survival

Median OS was significantly higher (*p* < 0.01) in patients treated with CE chemotherapy for PSC-NE carcinoma (14.4 months) compared to patients with primary adenocarcinoma with or without NE marker elevation (7.7 and 4.7 months, respectively, *p* = 0.47) (Figure 3A). However, as CE chemotherapy is used in routine practice (as in our series) as first-line treatment in patients with PSC-NE carcinoma and, on the contrary, at later stages of management in patients with primary adenocarcinoma, we also studied the OS since the diagnosis of metastatic disease. We thus confirmed that despite a greater benefit of CE chemotherapy at the time that this treatment was given to patients with PSC-NE carcinoma, the overall prognosis of these patients remained very poor (median OS since diagnosis of metastatic disease: 17 months) compared to patients treated for primary adenocarcinoma of the prostate with or without NE marker elevation (median OS since diagnosis of metastatic disease: 49.7 and 45 months, respectively) (Appendix A).

The same trends were observed for PFS: median PFS was significantly better (*p* < 0.01) in patients with PSC-NE carcinoma (7.9 months) compared to patients receiving CE chemotherapy in the context of later lines of treatment for primary adenocarcinoma with or without NE marker elevation (3.7 and 2.1 months, respectively) (Figure 3B).

Since patients with primary prostate adenocarcinoma received CE chemotherapy at different stages of their metastatic disease, we examined the PFS2/PFS1 ratio for each of them according to the presence or absence of NE marker elevation. Thus, we next evaluated part of the clinical benefit as measured by the percentage of patients with PFS on CE chemotherapy (PFS2), which was 1.3-fold higher than PFS prior to the systemic therapy (detailed in Appendix A) for metastatic disease (PFS1). Median PFS1 was 4.1 months, and median PFS2 was 2.3 months. The PFS2/PFS1 ratio was > 1.3 in 25.5% of patients (13/49). Individual PFS1 and PFS2 swimmer plots for patients treated with CE chemotherapy for primary adenocarcinoma of the prostate are presented in Figure 4. Of note, the proportion of patients with a PFS2/PFS1 ratio > 1.3 was higher in patients with NE marker elevation (43.8%) than in those without NE marker elevation (18.2%).

#### 3.2.6. Tolerance

The toxicity assessment is displayed in Table 2. Overall, 43 patients (62.3%) experienced grade 3–4 adverse events (AE) during CE chemotherapy. This proportion was not statistically different between the PSC-NE and the patients with primary adenocarcinoma. The most frequent significant AEs were related to hematological toxicity, with grade 3–4 anemia in 26 patients (37.7%), grade 3–4 neutropenia in 17 patients (24.6%) (including 6 patients (8.7%) presenting febrile neutropenia (4 of whom were patients who did not undergo any G-CSF prophylaxis)), and grade 3–4 thrombopenia in 9 patients (13%). Compared to patients treated for primary adenocarcinoma, hematological toxicity tended to be more frequent in patients treated with PSC-NE carcinoma (possibly due to a greater number of cycles of CE administered). Grade 3–4 nausea and/or vomiting were reported in seven patients (10.1%), and grade 3–4 creatinine elevation was observed only in two patients (2.9%). Grade 3–4 asthenia was more frequently recorded in patients treated for primary adenocarcinoma (41.2%) compared to patients treated for PSC-NE carcinoma (11.1%).

Four patients (one with PSC-NE carcinoma and three with primary adenocarcinoma) had to discontinue treatment permanently due to side effects. Finally, we recorded three treatment-related deaths (all due to septic shock), including two in patients who received G-CSF prophylaxis.

## 4. Discussion

In this retrospective study, we report efficacy and safety data from the use of a carboplatin/etoposide (CE) combination for the treatment of different subtypes of mPC. Our results clearly show that these different subtypes of disease need to be analyzed separately in order to assess the therapeutic value of this chemotherapy regimen.

Numerous chemotherapy agents have been tested and have shown clinical benefit in mPC, with mitoxantrone [17] first, followed by docetaxel [1] and cabazitaxel [2], with an OS benefit in patients with mCRPC. At the same time, the development of NHAs has revolutionized the management of mPC, improving the OS of patients at various stages of disease, both before and after chemotherapy. However, in current clinical practice, many patients have mCRPC that has become resistant to all chemotherapy and NHA treatments, even though they remain in good general condition. These patients require the continuation of a specific anti-tumor treatment. In this context of very advanced disease, and with a very limited level of evidence in terms of clinical benefit, the balance between the therapeutic/symptomatic efficacy and the toxicity of the chosen treatment must be very carefully assessed so as to avoid being deleterious for the patient.

While a majority of mCRPC remain AR-driven during tumor progression, it has now been shown that a subset of mPC is able to adapt under the therapeutic pressure of hormone therapy by becoming less AR dependent (and thus losing luminal cell markers such as PSA) and developing lineage plasticity. This includes the acquisition of biological and molecular characteristics usually encountered in anaplastic small-cell and/or NE tumors [12], forms usually treated with CE combination in first-line regimens. These poor-prognosis variants have no consensual definition [18] and may correspond to either clinically aggressive forms (associated with a high visceral metastatic burden) and/or biological forms (associated with the acquisition of NE markers and/or a loss of AR and PSA expression) [19] and/or molecular forms, with a frequently encountered genomic signature associating loss of TP53, Rb, and/or PTEN [20,21]. Most studies evaluating treatments for these aggressive forms, with or without NE differentiation, are old [8,10,11,22] and mostly used platinum-based chemotherapy regimens, often (but not exclusively) combined with etoposide. Platinum salts have long been tested in prostate cancers of all stages, but mainly in mCRPC [7]. Based on a phase I/II trial [9], the NCCN guidelines discuss the option of carboplatin + cabazitaxel chemotherapy for mCRPC patients with these poor prognostic criteria. However, the APCCC recommendations have not led to a consensus on the specific management of these poor-prognosis clinico-biological variants. Moreover, for a significant proportion of panelists, these aggressive variants should ultimately receive standard treatments for mCRPC [18]. Indeed, analyzing the results of these pioneering studies is complex, as they included patients with very different diseases (metastatic castrate-resistant adenocarcinoma with or without NE features or pure small-cell NE carcinoma), evaluated different chemotherapy regimens, and used different methods of response assessment. However, a phase 3 trial comparing satraplatin with prednisone showed a benefit in terms of PFS and pain control, but no benefit in terms of OS [23]. In these studies, the reported response rates are therefore very heterogeneous, ranging from 8 to 61%, and have a median OS from 8 to 19 months, depending on both the type of disease and the study period (taking into account other therapeutic options available at the time each study was carried out). For this reason, we aimed to report our series by separately analyzing three different populations (PSC-NE carcinomas and castration-resistant adenocarcinomas with or without biological signs of NE transformation) treated with the same combination chemotherapy regimen. For castration-resistant adenocarcinoma, we sought to include only patients who had previously received treatment with at least one NHA and a taxane, so as to report clinical results within the current mCRPC therapeutic landscape. Thus, our analyses clearly show that results mixing these three different populations should not be presented in clinical studies, since in clinical practice, the use of CE combination is earlier in the management of PSC-NE or mCRPC with NE features, thus probably biasing OS results between these patient groups (as demonstrated by our analysis of OS from the time of diagnosis of metastatic disease). Concerning the results of the CE combination for castration-resistant adenocarcinoma, one early study conducted in second-line treatment (after docetaxel failure) found a PSA response >50% in 23% of patients, associated with a clinical benefit (in terms of pain reduction) irrespective of the presence of detectable NE features [10]. Our study (with the same number of evaluable patients) shows lower PSA response rates and mainly concerns patients with NE marker elevation.

The results we report in terms of radiological response also support this view, with a low response rate (around 16%) in evaluable patients with primary adenocarcinoma without NE marker elevation. Furthermore, it should be noted that almost half of these patients had experienced rapid biological and/or clinical progression, necessitating the discontinuation of CE chemotherapy before the first radiological evaluation: the proportion of patients with mCRPC deriving objective benefit from CE at a late stage of their disease is therefore quite low (probably less than 10%) in our study. These trends, slightly more in favor of CE for tumors with NE marker elevation, were also observed when examining the PFS2/PFS1 ratio. Indeed, since PFS usually decreases over lines of treatment during the natural course of metastatic cancer [24], we chose to examine this endpoint in order to compare patients who received CE chemotherapy at different lines. A PFS2/PFS1 ratio > 1.3 was proposed to define a treatment benefit in such patients, and this ratio was used as a primary endpoint in the prospective MOSCATO 01 trial [25].

More recently, a better understanding of the biological heterogeneity of mPC in terms of drivers of tumor progression has revealed that a significant proportion (20–30%) of mCRPCs is characterized by DNA repair defects, making these tumors more sensitive to platinum salts [26] or to PARPi [27,28,29,30], especially in the case of BRCA2 mutations [31,32]. These abnormalities can be found in patients either at somatic or germline level, and some of them, such as deleterious BRCA1/2 mutations, have been associated with a clear benefit of PARPi in recent trials [6,33,34,35]. However, this therapeutic impact seems to be mainly derived from cases where a BRCA2 mutation is found, and whether other anomalies concerning genes involved in homologous recombination are also associated with the efficacy of DNA-damaging agents remains a matter of current debate. In a large international retrospective study conducted at 25 centers and including 508 patients with mCRPC treated with platinum-based chemotherapy (mostly in combination with a taxane or etoposide), the PSA > 50% response rates and radiological response appeared slightly higher (but not statistically significantly so) for the group of patients in whom an HRD gene abnormality (tumor or germline) was identified in molecular biology compared with patients in whom no abnormality was found (47.1% vs. 36.1% and 48.3% vs. 31.3%, respectively). In the remainder of the study population (who had not undergone genomic testing), the PSA > 50% and radiological response rates were 28.5% and 20.5%, respectively [36]. This study shows that the probability of response to platinum-based chemotherapy is greater in a population previously selected on the basis of homologous recombination anomalies (and here again, as with PARPi, especially in the case of BRCA2 [36] mutation), but that some patients without anomalies may also respond. Interpretation of these results is, however, rather difficult for the question asked by our study, as only a small percentage of patients in that study received a carboplatin/etoposide combination (18% of the cohort), and the presence of NE features was not specified. Moreover, as most patients received a platinum + taxane combination, the exact contribution of each chemotherapy family to the observed responses is complex to assess. Nevertheless, these results show that platinum-based chemotherapy may be a recommended treatment for patients with molecularly selected tumors, particularly in a setting where PARPi would be unavailable. The main limitation of our study concerns the absence of molecular screening for homologous recombination abnormalities, preventing us from verifying whether or not patients responding to the CE combination had such a tumor abnormality.

Regarding the combination of a platinum salt with etoposide, we report, as others have before [11], a toxicity profile that does not appear favorable to the systematic recommendation of this combination outside the specific case of PSC-NE carcinomas, for which alternative therapeutic options are limited.

## 5. Conclusions

PSC-NE carcinomas should not be mixed with adenocarcinoma-type mCRPCs when a CE combination is studied in mPC, since the prognosis of each entity, the efficacy of the combination, and the precocity of its introduction in the therapeutic strategy make the comparison of response rates and survival with adenocarcinoma completely illogical. For adenocarcinoma, the risk/benefit ratio of this regimen seems unfavorable due to poor response (especially if no NE marker elevation is detected) and high toxicity. The arrival of PARPi with an OS benefit for patients with mCRPC associated with a BRCA mutation now makes it essential to test all patients to guide management. In years to come, this could enable us to better specify whether a patient population can still really benefit from platinum salt chemotherapy, alone or in combination with etoposide in prostate cancer.

## Figures and Tables

**Figure 1 cancers-16-00280-f001:**
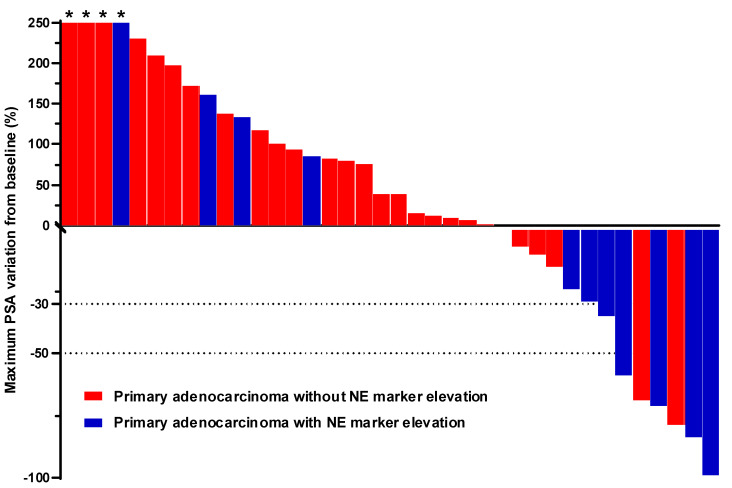
Best percent PSA change from baseline in patients with adenocarcinoma with or without NE marker elevation. For each patient, the best percent PSA change was calculated from the lowest PSA value at any time after baseline during treatment with CE. For this analysis, 38 patients presenting adenocarcinoma of the prostate with (blue bars) or without (red bars) NE marker elevation were included. Asterisks indicate a ≥ 250% increase in PSA values. Abbreviations: NE: neuroendocrine, CE: carboplatin/etoposide.

**Figure 2 cancers-16-00280-f002:**
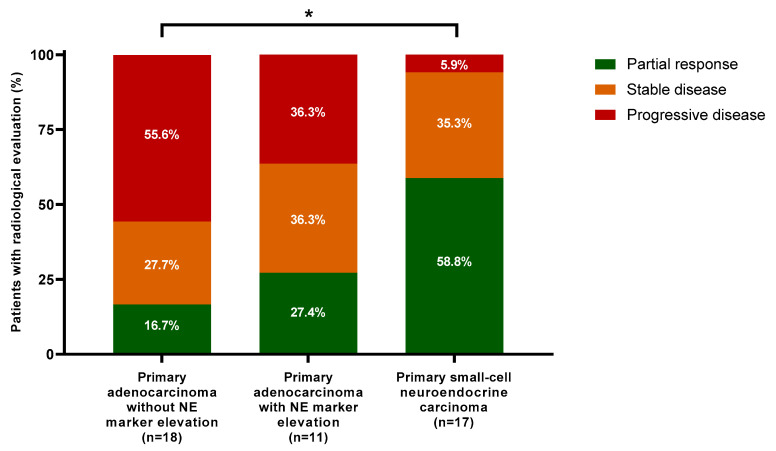
Best radiological response from baseline. For this analysis, 46 patients who had both baseline and evaluation CT and/or PET scans during treatment were included. No patient had complete response. Proportions were compared between groups using the chi-square test. Asterisks represent a statistically significant difference between groups (*p* < 0.05). Abbreviations: NE, neuroendocrine.

**Figure 3 cancers-16-00280-f003:**
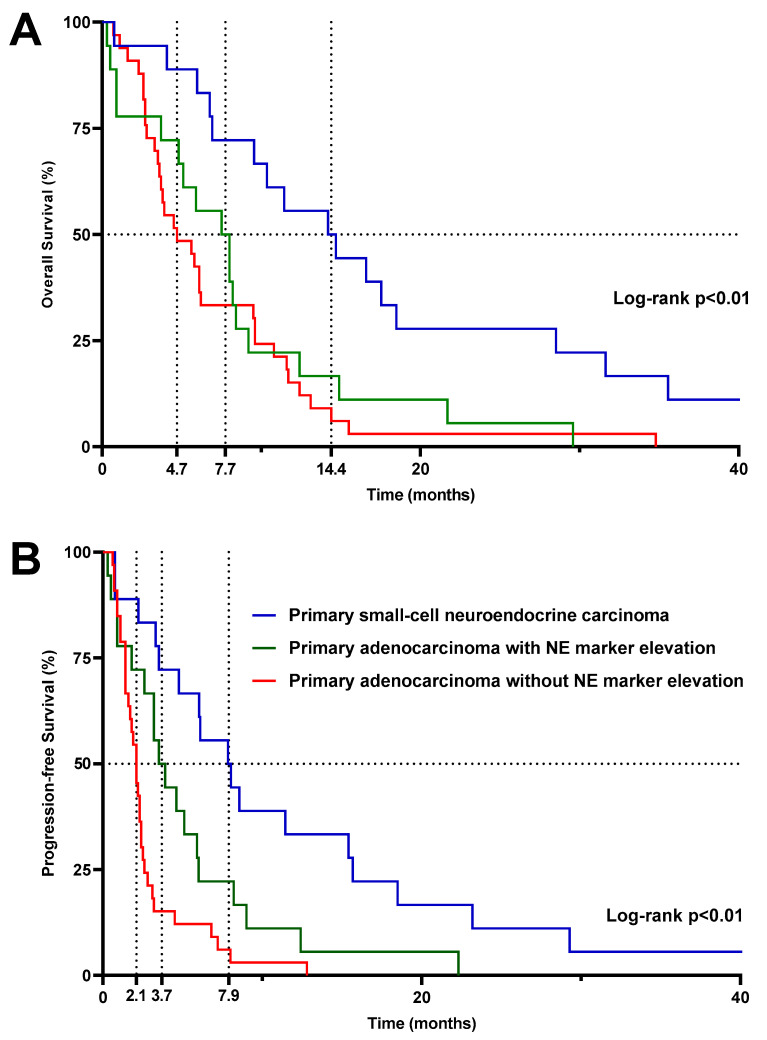
Kaplan–Meier estimates of overall survival (**A**) and progression-free survival (**B**) since the beginning of the carboplatin/etoposide combination. Dashed lines represent median survival. *p*-values shown on the graphs refer to comparisons of the three groups using the log-rank test in a univariate analysis. For statistical comparison of the “primary adenocarcinoma without NE marker elevation” and the “primary adenocarcinoma with NE marker elevation” groups using the log-rank test, *p* = 0.47 for overall survival and *p* = 0.02 for progression-free survival. Abbreviations: NE, neuroendocrine.

**Figure 4 cancers-16-00280-f004:**
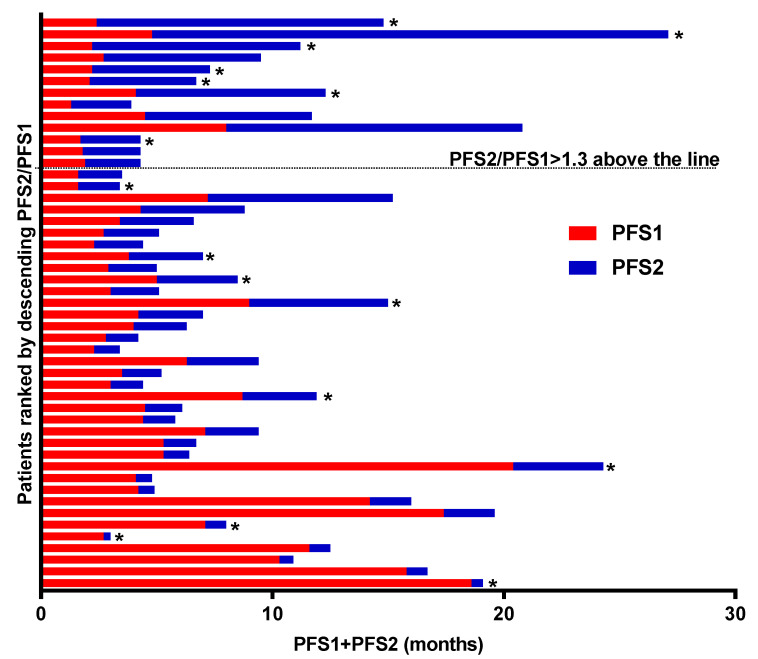
Individual PFS ratio. PFS2 and PFS1 represent the PFS interval associated with carboplatin/etoposide therapy (PFS2, blue bars) and with the last previous systemic therapy (PFS1, red bars), respectively. A total of 49 patients with primary adenocarcinoma with or without NE marker elevation were included in this analysis. Asterisks represent cases with NE marker elevation. Abbreviations: NE, neuroendocrine.

**Table 1 cancers-16-00280-t001:** Characteristics of patients at baseline and previous treatments.

	Overall Population	Primary Small-Cell Neuroendocrine Carcinoma	Primary Adenocarcinoma		
			No Elevation of NE Markers	Elevation of NE Markers	*p*-Value
** *N* **	69	18	33	18	
Mean age, years (SD)	69.4 (9.3)	67.2 (9.0)	71.6 (9.5)	67.4 (8.8)	0.12
ECOG status (%)					0.65
0–1	38 (55.1)	14 (77.8)	14 (42.4)	10 (55.6)	
>1	31 (44.9)	4 (22.2)	19 (57.6)	8 (44.4)	
Gleason score (%)					0.60
6 or 7	26 (37.7)	5 (27.8)	13 (39.4)	8 (44.4)	
8, 9 or 10	31 (44.9)	7 (38.9)	17 (51.5)	7 (38.9)	
Unknown	12 (17.4)	6 (33.3)	3 (9.1)	3 (16.7)	
Median PSA at start of carboplatin/etoposide ng/mL (Q1–Q3)	103.0 (7.5–422.0)	4.4 (0.8–12.9)	232.0 (103.0–681.0)	58.0 (6.2–267.0)	0.03
Median NSE at start of carboplatin/etoposide ng/mL (Q1–Q3)	23.4 (10.2–46.6)	50.3 (15.4–331.0)	9.25 (8.6–10.3)	26 (12.2–37.6)	0.03
Median chromogranin A at beginning of carboplatin/etoposide ng/mL (Q1–Q3)	161.5 (124.5–1204.3)	232.0 (127.0–612.8)		161.5 (126.5–1639.5)	
Metastatic sites (%)					0.72
Bones	60 (87.0)	12 (66.7)	31 (93.9)	17 (94.4)	
Lymph nodes	43 (62.3)	10 (55.6)	23 (69.7)	10 (55.6)	
Liver	24 (34.8)	9 (50.0)	9 (27.3)	6 (33.3)	
Lung	20 (29.0)	5 (27.8)	8 (24.2)	7 (38.9)	
Brain	4 (5.8)	1 (5.6)	2 (6.1)	1 (5.6)	
Other visceral	12 (17.4)	2 (11.1)	8 (24.2)	2 (11.1)	
None	4 (5.8)	1 (5.6)	1 (3.0)	2 (11.1)	
Median number of metastatic sites (Q1–Q3)	2 (2–3)	2 (1.25–2.75)	2 (2–3)	2 (2–3)	1
Median prior treatment lines (Q1–Q3)	3 (1–4)	0 (0–0)	4 (4–5)	3 (1–4)	0.01
Previous NHA (%)					<0.01
0	27 (39.1)	17 (94.4)	2 (6.1)	8 (44.4)	
1	13 (18.9)	1 (5.6)	8 (24.2)	4 (22.3)	
2	29 (42.0)	0 (0)	23 (69.7)	6 (33.3)	
Previous chemotherapies (%)					<0.01
0	17 (24.6)	15 (83.3)	0 (0)	2 (11.1)	
1	10 (14.5)	3 (16.7)	2 (6.1)	5 (27.8)	
2	25 (36.2)	0 (0)	20 (60.6)	5 (27.8)	
3	10 (14.5)	0 (0)	6 (18.2)	4 (22.3)	
4	3 (4.4)	0 (0)	2 (6.1)	1 (5.5)	
5 or more	4 (5.8)	0 (0)	3 (9.0)	1 (5.5)	
Patients previously treated by taxanes (%)					0.34
Docetaxel	49 (71.0)	2 (11.1)	33 (100)	16 (88.9)	
Cabazitaxel	38 (55.1)	0 (0)	30 (90.9)	8 (44.4)	
Rechallenge docetaxel	11 (15.9)	0 (0)	6 (18.2)	5 (27.8)	
Rechallenge cabazitaxel	3 (4.4)	0 (0)	2 (6.1)	1 (5.6)	

Statistical analyses were performed for continuous variables using Student’s *t*-test or Wilcoxon test after testing the normality of the distribution using the Shapiro–Wilk test. For categorical variables, Fisher’s exact test was performed. The displayed *p*-values account for the comparison of the “primary adenocarcinoma without NE marker elevation” and the “primary adenocarcinoma with NE marker elevation” groups in a univariate analysis. Abbreviations: NE: neuroendocrine. SD: standard deviation. Q1–Q3: quartile 1–3. NSE: neuron-specific enolase. NHA: next-generation hormonal agent.

**Table 2 cancers-16-00280-t002:** Treatment-related toxicity.

	Overall Population	Primary Small-Cell Neuroendocrine Carcinoma	Primary Adenocarcinoma with or without Elevated NE Markers
*N*	69	18	51
All-type toxicities (%)			
Grade 3–4	43 (62.3)	12 (66.7%)	31 (60.8%)
Grade 4	6 (8.7)	2 (11.1%)	4 (7.8%)
Treatment-related death	3 (4.3)	1 (5.6%)	2 (3.9%)
Grade 3–4 toxicity (%)			
Anemia	26 (37.7)	8 (44.4%)	18 (35.3%)
Neutropenia	17 (24.6)	7 (38.9%)	10 (19.6%)
Thrombopenia	9 (13.0)	5 (27.8%)	4 (7.8%)
Nausea and vomiting	7 (10.1)	3 (16.7%)	4 (7.8%)
Creatinine elevation	2 (2.9)	1 (5.6%)	1 (2.0%)
Asthenia	23 (33.3)	2 (11.1%)	21 (41.2%)
Febrile neutropenia (%)	6 (8.7)	3 (16.7%)	3 (5.88%)
Treatment interruption because of toxicity (%)	4 (5.8)	1 (5.6%)	3 (5.88%)

Abbreviations: NE, neuroendocrine.

## Data Availability

Data are available to readers promptly upon request from the corresponding author.

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
