# Peer review of "Carboplatin and Etoposide for the Treatment of Metastatic Prostate Cancer with or without Neuroendocrine Features: A French Single-Center Experience"

_cancers, 2024, doi:10.3390/cancers16020280_

Round 1

Reviewer 1 Report

Comments and Suggestions for Authors

In the manuscript entitled” Chemotherapy combining carboplatin and etoposide for the 2 treatment of metastatic castration-resistant or neuroendocrine 3 prostate cancer: a French single-center experience” by Baude et al, the authors have conducted a retrospective study focusing on prostate cancer patients who underwent treatment with carboplatin and etoposide. Mainly three different patient populations were studied: patients with PSC-NE carcinoma of the prostate, patients with adenocarcinoma (mCRPC) presented with biological signs of NE transformation/ elevation in NE markers and patients with adenocarcinoma (mCRPC) who had no signs of NE transformation (no NE marker elevation).

While the study provides valuable insights, the design of the study has some limitations.

The major concern is with the patient categorization. Patient information is not adequately provided and groups are not clearly described.

How are patients with PSC-NE carcinoma different from patients with adenocarcinoma (mCRPC) presented with  biological signs of NE transformation ?  Neuroendocrine differentiation in prostate cancer rarely occurs de novo and studies have shown that prolong treatment with androgen deprivation therapy(ADT) can promote neuroendocrine differentiation as well as elevation in NE markers in mCRPC patients. It is therefore essential to know whether the patients with primary small cell neuroendocrine carcinoma have received any prior treatment with ADT or hormonal agents before the carboplatin etoposide treatment.

While cisplatin or carboplatin in combination with etoposide stands as a standard treatment for poorly differentiated neuroendocrine cancers, its application in a phase II clinical study concerning patients with progressive metastatic castration-resistant prostate cancer (mCRPC) produced less favorable outcomes. This study revealed that the combination did not demonstrate a significant benefit-to-risk ratio, and it notably resulted in severe toxicity among the patients enrolled. Though the study highlighted clear differences in OS and PFS between patients with poorly differentiated neuroendocrine (PSC-NE) prostate cancer and the other two groups compared, there wasn't a substantial variance in toxicity levels observed among these groups. This raises an important consideration regarding the utility of this combination in PSC-NE patients. Authors should carefully weigh these findings when evaluating the appropriateness of recommending this regime for this group of patients.

The figures provided seem to lack adequate explanation and the legends accompanying them would greatly benefit from inclusion of additional details.   For e.g. Figure 1 does not provide sufficient information. At what points the PSA changes were measured.

 Similar information is needed for Figure 3.

The authors indicated  that a majority of the patients enrolled in the study exhibited bone metastasis, and approximately one-third presented with visceral metastasis. Among the 59 enrolled patients, there were 18 individuals categorized under the primary adenocarcinoma group, while the remaining 18 fell into the PSC-NE subgroup. However, the specific distribution of metastatic presentations within each group, including the number of patients and the extent of metastasis observed, was not delineated in the provided data.

The study findings revealed that there was no significant benefit observed among patients with adenocarcinoma (mCRPC) who exhibited biological indications of neuroendocrine (NE) transformation, characterized by elevated NE markers and also in patients with adenocarcinoma (mCRPC) who have no signs of NE transformation. Authors have mentioned that patients within the mCRPC subgroup had a history of extensive prior treatments but did not provide specific details regarding the types of agents administered or the duration of treatment received by these patients.

Reviewer 2 Report

Comments and Suggestions for Authors

The manuscript titled “Chemotherapy combining carboplatin and etoposide for the treatment of metastatic castration-resistant or neuroendocrine prostate cancer: a French single-center experience” aimed to investigate the benefit of carboplatin and etoposide combination therapy, especially in patients with heavily pre-treated adenocarcinoma. Authors revealed that the reports of clinical results of CE should not mix PSC-NE and adenocarcinoma and claim that in patients with PSC-NE, 58.8% has a radiological response to CE and median progression-free survival was 7.9 months. Authors concluded that in pretreated patients with adenocarcinoma, the benefit-risk ratio of CE seems unfavorable with poor response and high toxicity, irrespective of the elevation of neuroendocrine markers. The manuscript well written and is interesting.

However, there are few minor revisions as following:

1. title of the manuscript is too complex and unclear. Authors are encouraged to simplify it.

2.  Please provide the details of the statistical analysis done for each of the data analysis.

3. Please provide the details of the ethical clearance and institutional guidelines for utilizing human resources/data.
